# Effect of Isolated Scenting Process on the Aroma Quality of Osmanthus Longjing Tea

**DOI:** 10.3390/foods13182985

**Published:** 2024-09-20

**Authors:** Jianyong Zhang, Yuxiao Mao, Yongquan Xu, Zhihui Feng, Yuwan Wang, Jianxin Chen, Yun Zhao, Hongchun Cui, Junfeng Yin

**Affiliations:** 1Tea Research Institute, Chinese Academy of Agricultural Science, Hangzhou 310008, China; zjy5128@tricaas.com (J.Z.); xuyq@tricaas.com (Y.X.); fengzhihui@tricaas.com (Z.F.);; 2Hangzhou Academy of Agricultural Science, Hangzhou 310024, China

**Keywords:** osmanthus, Longjing tea, scenting process, aroma quality, metabolomics, flavor chemical, volatile compounds

## Abstract

Scenting is an important process for the formation of aroma quality in floral Longjing tea. There are differences in the aroma quality of osmanthus Longjing teas processed by different scenting processes. The efficient isolated scenting method was employed to process a new product of osmanthus Longjing tea in this study, and this was compared with the traditional scenting method. The volatile compounds of osmanthus Longjing tea were analyzed by a GC-MS instrument. In addition, the effects of scenting time and osmanthus consumption on the aroma quality of Longjing tea were studied. The results indicated that there were 67 kinds of volatile compounds in the osmanthus Longjing tea produced by the isolated scenting process (O-ISP), osmanthus Longjing tea produced by the traditional scenting process (O-TSP), and raw Longjing tea embryo (R), including alcohols, ketones, esters, aldehydes, olefins, acids, furans, and other aroma compounds. The proportions of alcohol compounds, ester compounds, aldehyde compounds, and ketone compounds in O-ISP were higher than in O-TSP and R. When the osmanthus consumption was increased, the relative contents of volatile aroma compounds gradually increased, which included the contents of trans-3,7-linalool oxide II, dehydrolinalool, linalool oxide III (furan type), linalool oxide IV (furan type), 2,6-Dimethyl cyclohexanol, isophytol, geraniol, 1-octene-3-alcohol, cis-2-pentenol, trans-3-hexenol, β-violet alcohol, 1-pentanol, benzyl alcohol, trans-p-2-menthene-1-alcohol, nerol, hexanol, terpineol, 6-epoxy-β-ionone, 4,2-butanone, 2,3-octanedione, methyl stearate, cis-3-hexenyl wasobutyrate, and dihydroanemone lactone. When the scenting time was increased, the relative contents of aroma compounds gradually increased, which included the contents of 2-phenylethanol, trans-3,7-linalool oxide I, trans-3,7-linalool oxide II, dehydrolinalool, isophytol, geraniol, trans-3-hexenol, β-ionol, benzyl alcohol, trans-p-2-menthene-1-ol, nerol, hexanol, terpineol, dihydroβ-ionone, α-ionone, and β-ionone,6,10. The isolated scenting process could achieve better aroma quality in terms of the floral fragrance, refreshing fragrance, and tender fragrance than the traditional scenting process. The isolated scenting process was suitable for processing osmanthus Longjing tea with high aroma quality. This study was hoped to provide a theoretical base for the formation mechanism and control of quality of osmanthus Longjing tea.

## 1. Introduction

Flower tea is a unique type of reprocessed tea in China, and it is also a very important type of reprocessed tea. It is made by mixing and fermenting various fragrant flowers with raw tea [1]. During the scenting process, a series of physical and chemical changes mainly include the release of the aroma of fresh flowers and the adsorption of aroma compounds by tea leaves, both making floral tea fresh, mellow, and refreshing as well as emanating the fragrant fragrance of flowers [2]. The main raw material for making scented tea is roasted green tea [3]. There are also a small number of stir-fried green teas, and some high-end premium green teas, while there are fewer oolong teas, black teas, and yellow teas scented with floral teas [4,5]. Longjing tea was a kind of world-famous roasted green tea from China [6,7]. However, Longjing tea was rarely used as a raw material for scented flower tea. The main reason was that traditional scented techniques could not achieve good aroma absorption effects. It was very necessary to innovate the process and organically integrate the flavor of Longjing tea for fresh flowers in order to create flower Longjing tea and improve its resource utilization and added value.

According to the different fragrant flowers used in scented flower teas, flower teas can be divided into jasmine flower tea, osmanthus tea, etc. [8,9,10,11]. Among them, jasmine tea has the largest proportion in the flower tea market [12]. However, the cultivation and flowering period of jasmine flowers are limited by certain regions and seasons, which leads to certain limitations in the production cost and expansion of jasmine tea production. The osmanthus is a plant of the oleaceae family and is one of the top ten traditional famous flowers in China [13]. The osmanthus with strong fragrance usually blooms in September and October. The osmanthus could be used to make osmanthus black tea and osmanthus green tea [14,15]. In recent years, flavonoids, polysaccharides, and triterpenoids of osmanthus have been found to have antibacterial activity [16,17], anti-inflammatory activity [18], antitumor activity [19], antioxidant activity [20,21], hypoglycemic activity [22], etc. The osmanthus has a wide range of cultivation and is widely planted in most parts of the world [23]. The osmanthus is an ideal flower material for making scented tea.

Flower scenting is one of the most important steps in flower tea processing and could be mainly divided into traditional flower scenting and isolated flower scenting [24]. The method of isolated and scenting flowers involves separating fresh flowers and tea and placing them in different compartments. By using air separation and fragrance treatment, the time of screening and removing flowers is reduced, which not only improves the effect of flower scenting but also reduces labor consumption and shortens the production cycle. At present, isolated scented flowers have been applied in the processing of jasmine tea [25,26]. However, it was not applied in the processing of osmanthus Longjing tea. The scenting process of osmanthus tea could be divided into seven steps including fresh flower harvesting and processing, tea processing, scenting and mixing, continuous scenting with flowers, reheating with flowers, jacquard and uniform stacking, and packing [27]. Aroma is the main factor affecting the comprehensive quality of tea, accounting for up to 35% in the sensory evaluation of floral tea [28]. The scenting processing could significantly improve the flavor quality of tea. The consumption of flowers and the scenting time are two important factors that affect the quality of floral tea. If the scenting time is too long and the number of flowers is too high, the tea aroma and taste will be bad, and there will be more wastage of flowers. If the time for scenting is not enough and the number of flowers is too small, it will lead to a low aroma absorption rate, resulting in a weak aroma and poor flavor for the flower tea. It is of great significance for improving the quality of osmanthus Longjing tea to make clear the consumption of flowers and the scenting time. The sensory evaluation and LC-MS/MS-based metabolomics were important technical methods for analyzing the quality chemistry of tea and the flavor quality of flower tea [29,30,31].

At present, more research is focused on the study of various types of flower teas such as jasmine tea or osmanthus black tea [32,33,34]. Osmanthus tea is mainly processed by traditional scenting methods [35], which have poorer aroma absorption effects [8,11]. According to the equation describing seven key volatile compounds and the quality of jasmine tea, the optimal scenting technology was obtained: the flowers consumption was 65–78%, scenting time was 15–17 h, and scenting temperature was 35–40 °C [32]. When the proportion of flower dosage was between 69% and 115%, the contents of aroma compounds in jasmine tea increased with an increase in flower dosage by the wet scenting process [33]. The 28 aroma compounds had been founded in Dangui-flower black tea, the contents of which were significantly higher than the original black tea leaves [34]. However, research on osmanthus Longjing green tea was still relatively weak, especially in terms of the influence of innovative scenting processes on the aroma quality of Longjing scenting tea. It was very necessary to study the effect of innovative scenting technology on the processing of osmanthus Longjing tea.

An innovative isolated scenting method to process osmanthus Longjing tea was employed in this study. The influence of different process parameters of isolated scenting on the aroma quality of osmanthus Longjing tea was clarified. Longjing tea and fresh osmanthus as raw materials were used to compare the effects of the traditional scenting method and isolated scenting method on the flavor and quality of osmanthus Longjing tea. The analysis methods of HS-SPME, GC-MS, and metabolomics were used to compare the differences in aroma compounds under different osmanthus consumption and scenting times. The effects of osmanthus consumption and scenting times on the aroma and quality of osmanthus Longjing tea were hoped to be studied. A theoretical basis for innovation in the processing technology and flavor quality control of osmanthus Longjing tea was hoped to be provided.

## 2. Materials and Methods

### 2.1. Materials and Equipment

The Longjing tea was obtained from tea research institute of Hangzhou Academy of Agricultural Sciences, Zhejiang Province, China.

Normal alkanes (C8–C40, ≥99%, chromatographically pure), N-alkane (≥99%, chromatographically pure), ethyl decanoate (≥99%, chromatographically pure), and ethanol (≥99%, chromatographically pure) were purchased from Beijing Bailingwei Technology Co., Ltd. (Beijing, China). Deionized water was purchased from Hangzhou Wahaha Group Co., Ltd. (Hangzhou, China). Dimethoxysilane mixing rod (length: 10 mm, thickness: 0.5 mm, volume: 24 μL) was purchased from GMBH Corporation (Dusseldorf, North Wales, Germany). Tube mill control was purchased from IKA Corporation (Staufen, Germany). Analytical electronic balance (SQP type) was purchased from Sedolis Scientific Instruments Co., Ltd. (Beijing, China). SP200-2T multi-channel magnetic stirrer was purchased from Hangzhou Mio Instrument Co., Ltd. (Hangzhou, China).

### 2.2. Two Scenting Process Treatments of Osmanthus Longjing Tea

The traditional scenting process of osmanthus Longjing tea is shown in Figure 1. Fifteen percent of the osmanthus consumption was mixed with Longjing tea for 16 h. In order to achieve better heat dissipation effect and flower absorption effect, the mixture of Longjing tea and osmanthus was flipped over during the traditional scenting process. Then, the flower residue was sieved out. Afterwards, it was subjected to an 80 °C baking process. The osmanthus Longjing tea prepared by the traditional scenting process was obtained.

The isolated scenting process of osmanthus Longjing tea is shown in Figure 2. Fresh osmanthus and Longjing tea were stored separately in different storage tanks. The bottom of each storage tank was hollowed out, which facilitated the release of osmanthus fragrance and absorption by Longjing tea. The quantity of fresh osmanthus used in the isolated scenting process was also 15%, which was the same as the quantity of fresh osmanthus used in traditional scenting method.

The fresh osmanthus flowers were added to Longjing tea according to the proportion of 15% and mixed well with Longjing tea for 16 h. In order to achieve better heat dissipation effect and flower absorption effect, the mixture of Longjing tea and osmanthus was flipped over during the traditional scenting process. Then, the flower residue was sieved out. Afterwards, it was subjected to an 80 °C baking process. The osmanthus Longjing tea prepared by the traditional scenting process was obtained.

### 2.3. Effect of Osmanthus Consumption on Volatile Compounds of Osmanthus Longjing Tea

The Jingui osmanthus and Longjing tea were used as raw materials to process osmanthus Longjing tea. The isolated scenting process was the same as that of the treatment in Section 2.2. The levels of osmanthus consumption were designed for 12%, 15%, 18%, and 21%. The volatile compounds of osmanthus Longjing tea with different osmanthus consumptions were analyzed by GC-MS.

### 2.4. Effect of Scenting Time on Aroma of Osmanthus Longjing Tea Prepared by Isolated Scenting Process

The Jingui osmanthus and Longjing tea were used as raw materials to process osmanthus Longjing tea. The isolation scenting process was the same as that of the treatment in Section 2.2. The levels of scenting time were designed for 10 h, 13 h, 16 h, and 19 h. The volatile compounds of osmanthus Longjing tea for different scenting times were analyzed by GC-MS.

### 2.5. Preparation of Longjing Tea Infusions and Osmanthus Longjing Tea Infusions

3 g Longjing tea and osmanthus Longjing tea were separately soaked in 150 mL hot water at 100 °C for 4 min to obtain tea infusions. The Longjing tea infusions and osmanthus Longjing tea infusions were filtered with 200 mesh filter and quickly cooled to 20 °C using an ice bath [8]. The above Longjing tea infusions and osmanthus Longjing tea infusions were preserved for sensory evaluation and aroma compound analysis.

### 2.6. Sensory Evaluation of Tea Samples

Qualitative and quantitative descriptive methods were employed to evaluate the aroma quality characteristics and intensity of osmanthus Longjing tea and of raw Longjing tea, which was slightly modified [8,32]. The qualitative descriptions of tea infusions aroma included ‘floral’, ‘fullness’, ‘durability’, and ‘fresh’. The floral fragrance mainly referred to the smell of tea soup brewed with hot water from scented tea [2]. Floral fragrance was a very important quality parameter for osmanthus Longjing tea, specifically describing the osmanthus fragrance in osmanthus Longjing tea. Fullness referred to the total amount of aroma and sensory impact. Durability, also known as fragrance retention, referred to the time limit for the aroma of tea soup to remain in a sensory evaluation environment. Freshness referred to the degree of preservation and freshness of the aroma characteristics of tea soup. The quantitative description of the intensity of tea infusion aromas was evaluated by a nine-point representation methods, which included 0 for ‘none’, 1–3 for ‘weak aroma’, 4–6 for ‘obvious aroma’, and 7–9 for ‘strong aroma’. The score represented the intensities of the aroma attributes of tea. The higher the score was, the better the aroma intensities were and the higher the overall aroma quality of tea was.

A sensory panel of five expert panelists was recruited to evaluate tea aroma quality. The expert panelists had had more than five years of tea sensory evaluation experience. The five sensory evaluation experts were all senior tea appraisers (Grade I) certified by the Ministry of Human Resources and Social Security of China, including three women and two men. Five expert panelists smelled the tea infusion aroma fragrances one by one. The qualitative and quantitative descriptive evaluation results were recorded. The tea infusions were evaluated by expert panelists. The average score was used to represent the intensity of tea aroma, referring to the “GB/T 23776-2018 Methods for Sensory Evaluation of Tea” released by China [36]. This study developed an experimental plan for sensory evaluation of flower tea that complied with Chinese national laws and did not require ethical approval.

### 2.7. Analysis of Volatile Compounds in Tea Samples

#### 2.7.1. Solid-Phase Microextraction of Volatile Compounds from Tea Samples

The volatile compounds in tea samples contribute greatly to the aroma quality of tea. The extraction effects of volatile compounds by different extraction methods were different. Solid-phase microextraction (SPME) is a kind of method with high extraction efficiency and is widely used in the extraction of volatile compounds in tea [5,27]. The volatiles of Longjing tea and osmanthus Longjing tea were extracted by the SPME method with slight modification [11]. The 0.6 g tea samples were accurately weighed and sealed in 15 mL glass vial (Agilent, Santa Clara, CA, USA). The 10 mg/L ethyl decanoate as internal standard was added into the glass vial. Then, 5 mL of boiling deionized water was added into the above glass vial sequentially. The glass vial was immediately placed on a water bath at 60 °C and equilibrated for 3 min. The PDMS/DVB (50/30 μm) coating fiber (Supelco, Bellefonte, PA, USA) was used to absorb the head-space volatiles for 65 min at 60 °C and had been aged via gas chromatography at 250 °C for 5 min in advance. Then, the coating fiber was inserted into the inlet of the gas chromatography instrument and was desorbed at 250 °C for 5 min for data collection and analysis. The tea samples were analyzed three times.

#### 2.7.2. Analysis and Confirm of Volatile Compounds by GC-MS

Aroma was an important quality of Longjing tea and osmanthus Longjing tea. It was necessary to analyze the volatile compounds of tea samples by GC. The GC-MS method with slight modification was employed to analysis the volatiles of Longjing tea and osmanthus Longjing tea [4,8]. The Agilent 6890 GC system interfaced with the 5975 MSD ion trap MS (Agilent, Santa Clara, CA, USA) was employed to identify volatile compounds of tea samples. The HP-5 MS capillary column (30 m × 250 μm × 0.25 μm, Agilent, Santa Clara, CA, USA) was used in the split-less injection mode with a high-purity helium (99.999%) flow at 1.2 mL/min with a split ratio of 6:1. The inlet temperature of GC was 250 °C mode with high-purity helium (99.999%) as carrier gas at 1.0 mL/min. The following temperature program was used: initial temperature was 40 °C for 2 min; this was increased to 85 °C at a rate of 2 °C/min (held for 2 min), increased to 180 °C at a rate of 2.5 °C/min (held for 2 min), and increased to 230 °C at a rate of 10 °C/min (held for 2 min). The mass spectrometry analysis conditions were as follows: energy of 70 eV in electron ionization mode, mass scan range of 40–400 m/z, ion source interface temperature of 230 °C, and solvent delay time of 3.5 min.

The volatile compounds were retrieved and confirmed by the National Institute of Standards and Technology 17 (NIST 17) standard library. When the matching index of volatile compounds was more than 80%, these compounds were confirmed. The retention index (RI) was used to identify aroma compounds. According to the retention time of the compounds to be tested, the retention index (RI) was calculated and identified in the literature and MS database. According to the concentration of internal standard and percentage of peak area, the internal standard method was used to calculate the relative contents of aroma compounds.

### 2.8. Statistical Method

All experiments were conducted in triplicates. SIMCA 14.1 (Umetrics AB, Umea, Sweden) was employed for multivariate analysis. IBM SPSS software (version 23.0, SPSS Inc., Chicago, IL, USA) was employed for data analysis using one-way analysis of variance (*p* < 0.05) and significance testing.

Principal Component Analysis (PCA) was used to analysis the overall distribution among the tea samples and the stability of the whole process. Orthogonal partial least-squares-discriminant analysis (ENDOPLASM) was used to disgusting the different metabolites among groups. The Variable Importance of Projection (VIP) values were used to rank the overall contribution of each variable to group discrimination, which came from the OPLS-DA. Differential metabolites were selected with VIP values greater than 1.0 and *p*-values less than 0.05. The cross-validation model of 200 Response Permutation Testing (RPT) was used to evaluate the robustness of the model, which was used to prevent over-fitting.

## 3. Results and Discussion

### 3.1. Sensory Evaluation of Osmanthus Longjing Tea for Two Scenting Processes

There were differences in the aroma types of the osmanthus Longjing teas made by different scenting processes (Figure 3). R is a control treatment. The floral fragrance score, refreshing fragrance score, and tender fragrance score of R (raw Longjing tea embryo) were lower than those of O-TSP and O-ISP, especially in the case of the floral fragrance score. The chestnut fragrance score of R (raw Longjing tea embryo) was comparable to those of O-TSP and O-ISP. The floral fragrance score, refreshing fragrance score, and tender fragrance score of O-ISP were higher than those of O-TSP. The traditional scenting process enhanced the floral aroma of tea [8,10,25], but there was few reports on improving its clear and tender fragrance. It could be inferred that scenting with osmanthus significantly improved the floral fragrance, refreshing fragrance, and tender fragrance. The isolated scenting process could achieve a better aroma quality in terms of floral fragrance, refreshing fragrance, and tender fragrance than the traditional scenting process. Therefore, the isolated scenting process was suitable for processing osmanthus Longjing tea with high aroma quality.

### 3.2. Analysis of Volatile Compounds in Two Scenting Processes of Osmanthus Longjing Tea 

The aroma compounds of R, O-TSP, and O-ISP were analyzed by GC-MS. There were 67 kinds of common aroma compounds in the three samples, including alcohols, ketones, esters, aldehydes, olefins, acids, furans, and other aroma compounds (Figure 4). There were no significant differences in the proportions of pyrrole compounds, acidic compounds, and furan compounds among R, O-TSP, and O-ISP. The pyrrole compounds, acidic compounds, and furan compounds were usually the main material bases of chestnut fragrance [2]. That was to say that the differences in the chestnut fragrance quality among the above three types of tea would be relatively small. This was consistent with the evaluation results on the aroma sensory quality of the above three kinds of tea samples.

The proportions of alcohol compounds, ester compounds, aldehyde compounds, and ketone compounds of O-ISP were higher than those of O-TSP and R. The olefin compound proportions of O-ISP and O-TSP were higher than those of R. There was no significant difference in olefin compound proportions between O-ISP and O-TSP. The ester compounds and aldehyde compounds were usually the main material bases of floral fragrance [3]. The ketone compounds were usually the main material bases of tender fragrance [3]. The alcohol compounds and olefin compounds were usually the main material bases of refreshing fragrance [3]. It could be concluded that the flower fragrance, refreshing fragrance, and tender fragrance flavor qualities of Longjing tea were significantly improved by scenting with osmanthus, especially after the isolated scenting process [8,11]. The osmanthus Longjing tea contained high contents of aroma compounds such as phytol, β-ionone, hexadecanoic acid, butylated hydroxytoluene, linolenic acid, acetic acid phytol ester, (E)-furan type linalool oxide 2, and methyl hexadecanoate. However, the volatile compounds with higher contents in the Longjing tea were phytol, N-methylpyrrolidone, and methyl hexadecanoate. This indicated that the scenting process had significantly changed the composition characteristics of the volatile compounds in the raw Longjing tea, introducing the volatile compounds of osmanthus, which helped to form the floral characteristics of osmanthus Longjing tea.

As show in Figure 5, cluster analysis was carried out on the aroma compounds of R, O-ISP, and O-TSP. Compared with R and O-TSP, the up-regulated aroma compounds of O-ISP included 2-phenylethanol, trans-3,7-linalool oxide I, trans-3,7-linalool oxide II, isophytol, geraniol, trans-3-hexenol, β-ionol, 1-pentanol, benzyl alcohol, trans-p-2-menthene-1-ol, nerol, hexanol, terpineol, 3,5-octadiene-2-one, trans, trans. There were 28 kinds of 14-trimethyl-2-pentadecenone, acetophenone, methyl 12-octadecenoate, γ-decanolide, erythritol isovalerate, β-cyclocitral, 2-methylbutyraldehyde, valeraldehyde, and limonene. These up-regulated aroma compounds were the key compounds in osmanthus Longjing tea processed by the isolated scenting process and yet were the main contributors to the flavor quality of flowers fragrance, and tender fragrance. As can be seen from Table 1, the content of these compounds in osmanthus Longjing tea (O-ISP) reached 106.33 ± 0.85 μg/g, which was 22.2% higher than in the traditional-scenting-process Longjing tea (O-TSP, 82.75 ± 0.39 μg/g) and raw Longjing tea (R, 70.95 ± 0.27 μg/g). This indicated that some volatile compounds had been transferred from osmanthus Longjing tea to the raw Longjing tea through adsorption desorption, which helped give the raw Longjin tea a floral aroma. The down-regulated aroma compounds of O-ISP included 2,6-dimethylcyclohexanol, linalool I (furan type), linalool II (furan type), linalool III (furan type), linalool IV (furan type), dehydrolinalool, 1-octene-3-alcohol, cis-2-pentenol, methyl palmitate, and stearic acid, which also affected the aroma quality of the osmanthus Longjing tea. The osmanthus Longjing tea contained high contents of compounds such as phytol, β-ionone, hexadecanoic acid, butylated hydroxytoluene, linolenic acid, acetic acid phytol ester, (E)-furan type linalool oxide 2, and methyl hexadecanoate. However, the volatile compounds with higher contents in the raw Longjin tea were phytol, N-methylpyrrolidone, and methyl hexadecanoate. This indicated that the scenting process had significantly changed the composition characteristics of the volatile compounds in the raw Longjing tea, introducing the volatile compounds of osmanthus, which was conducive to form the floral characteristics of osmanthus Longjing tea.

### 3.3. Volatile Compound Analysis of Osmanthus Longjing Tea Prepared by Isolated Scenting Process with Different Consumptions of Osmanthus

The above results showed that osmanthus Longjing processed by the isolated scenting process had better aroma quality than the osmanthus Longjing processed by the traditional scenting process. In order to understand the influence of osmanthus consumption on the aroma quality of osmanthus Longjing tea prepared by the isolated scenting process, an orthogonal partial least-squares-discriminant analysis (OPLS-DA) was used to analyze the characteristic aroma compounds. A satisfying variance explanatory ability (R^2^X = 0.970, R^2^Y = 0.997) and high predictive power (Q^2^ = 0.985) of the OPLS-DA model was obtained (Figure 6A). As shown in Figure 6B, osmanthus Longjing teas with different consumptions of osmanthus were well distinguished on the score chart. The quality cross-validation of 200 permutation tests was subsequently employed to evaluation the reliability of the OPLS-DA model (Figure 6B). The fitting index was R^2^ = (0.0, 0.062); Q^2^ = (0.0, −0.856). The intercept value of Q^2^ was lower than zero, which indicated that there was no over-fitting situation in the OPLS-DA model. The OPLS-DA model was proved to have reliability.

An OPLS-DA regression coefficient plot for the aroma compounds of osmanthus Longjing treated with different consumptions of osmanthus was obtained (Figure 6C). The coefficient was positive for aroma compounds, which indicated that there was a significant positive correlation between these aroma compounds and the osmanthus consumption. The higher the correlation coefficient is, the stronger the correlation is. The aroma compounds that positively correlated with the osmanthus consumption were methyl palmitate, β-Cyclocitral, leaf alcohol isovalerate, linalool oxide I (furan type), 2-n-amyl furan, decylic acid, caprylic acid, enanthic acid, 1-ethyl-2-formyl pyrrole, pelargonic acid, geranic acid, linalool oxide II (furan type), and methyl 12-octadecenoate.

In order to further elucidate the differences in the contribution rates of aroma compounds in different consumptions of osmanthus, 16 crucial differential compounds were screened based on the criteria of *p* < 0.05 and variable importance in the projection (VIP) > 1 (Figure 6D). The aroma compounds with VIP > 1 were methyl palmitate, Linalool oxide I (furan type), methyl 12-octadecenoate, β-Cyclocitral, linalool oxide III (furan type), trans-3,7-linalool oxide I, leaf alcohol isovalerate, γ-decanolide, α -ionone, linalool oxide II (furan type), methyl linoleate, 2,2,6-tri-methylcyclohexane ketone, β-ionone, 3,5-Octadiene-2-one, trans-3,5-octadiene-2-one, and trans-3,7-linalool oxide II in descending order. Heterocyclic compounds were mainly produced by the maillard reaction during the heating and processing of tea. The ester aroma compounds were mainly related to the scenting and fatty acid metabolism of tea, which can endow osmanthus tea with fruity fragrance characteristics. The β-ionone, linalool, myrcene, and α-ionone were the important compounds in the formation of floral and fruity aromas in osmanthus tea and were also the main aroma compounds of osmanthus.

Cluster analysis was conducted on the aroma compounds of osmanthus Longjing raw tea with different consumptions of osmanthus (Figure 7). When the osmanthus consumption was increased, the relative contents of aroma compounds gradually increased; these compounds included trans-3,7-linalool oxide II, dehydrolinalool, linalool oxide III (furan type), linalool oxide IV (furan type), 2,6-Dimethyl cyclohexanol, isophytol, geraniol, 1-octene-3-alcohol, cis-2-pentenol, trans-3-hexenol, β-violet alcohol, 1-pentanol, benzyl alcohol, trans-p-2-menthene-1-alcohol, nerol, hexanol, terpineol, etc. 6-epoxy-β-ionone, 4-(2,2-dimethyl-6-methylene cyclohexyl)-2-butanone, 2,3-octanedione, methyl stearate, cis-3-hexenyl wasobutyrate, and dihydroanemone lactone. When the osmanthus consumption was decreased, the relative contents of aroma compounds gradually decreased: linalool oxide I (furan type), linalool oxide II (furan type), pentaerythritol isovalerate, methyl linoleate, γ-decanolide, methyl palmitate, β-cyclocitral, decanal,2-methyl-2-pentenal, lauric acid, geraniol, and 1-ethyl-2.

### 3.4. Volatile Compound Analysis of Osmanthus Longjing Tea Prepared by Isolated Scenting Process with Different Isolated Scenting Times

The scenting time was another important factor affecting the aroma quality of osmanthus Longjing tea prepared by the isolated scenting process. OPLS-DA was employed to analyze the characteristic aroma compounds with different scenting time treatment (Figure 8A). The variance fitting indexes were R^2^X = 0.946 and R^2^Y = 0.943, which indicated that the variance of the aroma compound quantities with different scenting time treatments was good. The predictive value was high (Q^2^ = 0.900), which indicated that the OPLS-DA model fit well. As shown in Figure 8B, the OPLS-DA model was proved to have reliability by the quality cross-validation of 200 permutation tests (Figure 8B). The permutation index was R^2^ = (0.0, 0.215); Q^2^ = (0.0, −0.763). The intersection point of the regression line and vertical line was lower than zero. These demonstrated no over-fitting situation in the OPLS-DA model.

An OPLS-DA regression coefficient plot for the aroma compounds of osmanthus Longjing treated with different scenting times was obtained (Figure 8C). The coefficient was positive for aroma compounds, indicating that there was a significant positive correlation between these aroma compounds and the scenting time. The higher the correlation coefficient is, the stronger the correlation is. The aroma compounds positively correlated with different scenting times were decanal, linalool oxide IV (furan type), dihydroanemone lactone, 6-methyl-5-heptene-2-one, 1-pentanol, 2-n-amyl furan, 1,4-eicosadiene, 1-ethyl-2-formyl pyrrole, lauric acid, decylic acid, caprylic acid, cis-2-pentenol, 1-octene-3-ol, geranic acid, β-myrcene, pelargonic acid, 2-methyl-2- pentenal, octadecene, limonene, enanthic acid, hexyl alcohol, β-Cyclocitral, 2,3-octanedion,α -ionone, trans-3,7-linalool oxide II, dihydroactinidiolide, terpineol, 6,10,14-trimethyl-2-pentadecanone, β-ionone, 2,2,6-tri-methylcyclohexane ketone, methyl tetradecanoate, acetyl benzene, methyl linoleate, linalool oxide III (furan type), methyl palmitate, 5,6-epoxy-β-ionone, and γ-decanolide.

In order to further elucidate the differences in the contribution rates of aroma compounds of different scenting times, 26 crucial differential compounds were screened based on the criteria of *p* < 0.05 and variable importance in the projection (VIP) > 1 (Figure 8D). Aroma compounds with VIP > 1 were linalool oxide IV (furan type), trans-3,5-octadiene-2-one, dihydroβ-ionone, α-cyclic citral, farniketone, safranal, methyl linolenate, 1-pentene-3-ol, trans-3,7-linalool oxide II, 2-phenylethanol, 6-methyl-5-heptene-2-one, leaf alcohol isovalerate, hexyl alcohol, 2,6-dimethylcyclohexanol, Linalool oxide I (furan type), decanal, 2,3-octanedione, β-violet alcohol, cis-3-hexenyl isobutyrate, methyl 12-octadecenoate, trans-3-hexenol, α-ionone, trans-p-2-menthene-1-ol, linalool oxide II (furan type), geraniol, and terpineol in descending order. Alpha ionone, dehydrolinalool, and phenylacetaldehyde endow osmanthus tea with sweet and honey aroma characteristics; Laurene, linalool, and citral exhibit rich floral and fruity aromas; Lauranol, ethyl caproate, and other compounds exhibit aromas such as fat and milk; 2,3-diethyl-5-methylpyrazine and other aromatic compounds present a baking aroma, which makes the fragrance of osmanthus tea more rich and intense.

Cluster analysis was conducted on the aroma compounds of osmanthus Longjing raw tea with different scenting times (Figure 9). When the scenting time was increased, the relative contents of aroma compounds gradually increased; these compounds included 2-phenylethanol, trans-3,7-linalool oxide I, trans-3, 7-linalool oxide II, dehydrolinalool, isophytol, geraniol, trans-3-hexenol, β-ionol, benzyl alcohol, trans-p-2-menthene-1-ol, nerol, hexanol, terpineol, dihydroβ-ionone, α-ionone, and β-ionone, 6, 10. When the scenting time was decreased, 1-octene-3-ol, cis-2-pentenol, 2,6-Dimethyl cyclohexanol, linalool I (furan type), linalool II (furan type), linalool III (furan type), linalool IV (furan type), 1-pentanol, methyl linoleate, γ-decanolide, methyl palmitate, dihydroactinidiolide, dihydroanemone lactone, methyl myristate, methyl linolenate, and decanal were decreased.

## 4. Conclusions

Scenting is a very important processing technology for scented tea. The changes in aroma components in the processing of scented tea have always constituted a hot spot of concern. The scenting process, the osmanthus consumption, and the scenting time had a great influence on the flavor and quality of osmanthus Longjing tea. The flavor of osmanthus Longjing tea processed by the isolated scenting method was better than that processed by the traditional scenting method, especially in terms of the floral fragrance and tender fragrance of the aroma quality. The contents of alcohols, esters, aldehydes, and ketones in osmanthus Longjing tea processed by the isolated scenting method were higher than those of osmanthus Longjing tea processed by the traditional scenting method. In the isolated scenting process, with an increase in the osmanthus consumption and the scenting time, the contents of alcohol compounds, ester compounds, and ketone compounds also increased. Therefore, alcohols, esters, and aldehydes were the important volatile material bases for the formation of the characteristic aroma quality of osmanthus Longjing tea. These three kinds of volatile substances might be used as important evaluation indexes for the quality of an isolation processing technology. The results of this study provide a new theoretical basis for the formation mechanism of flavor quality and the innovation of processing technology for osmanthus Longjing tea. A new insight for research on the scenting processing technology for osmanthus Longjing tea has also been provided. In future studies, the influences of different kinds of osmanthus fragrans and different kinds of tea embryos on the transformation and formation mechanism of non-volatile and volatile metabolites of osmanthus Longjing tea are hoped to be explored. The potential influences of temperature and humidity changes on the flavor characteristics of osmanthus Longjing tea during the scenting process deserve in-depth study.

## Figures and Tables

**Figure 1 foods-13-02985-f001:**
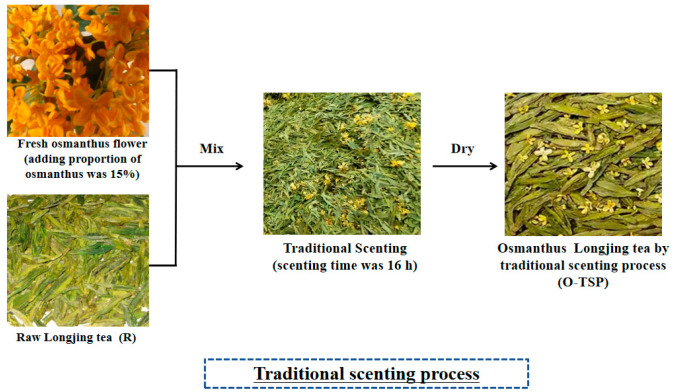
Traditional scenting process of osmanthus Longjing tea.

**Figure 2 foods-13-02985-f002:**
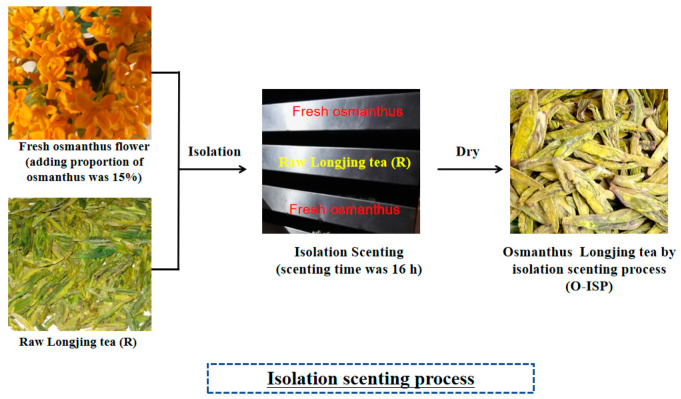
Isolated scenting process for osmanthus Longjing tea.

**Figure 3 foods-13-02985-f003:**
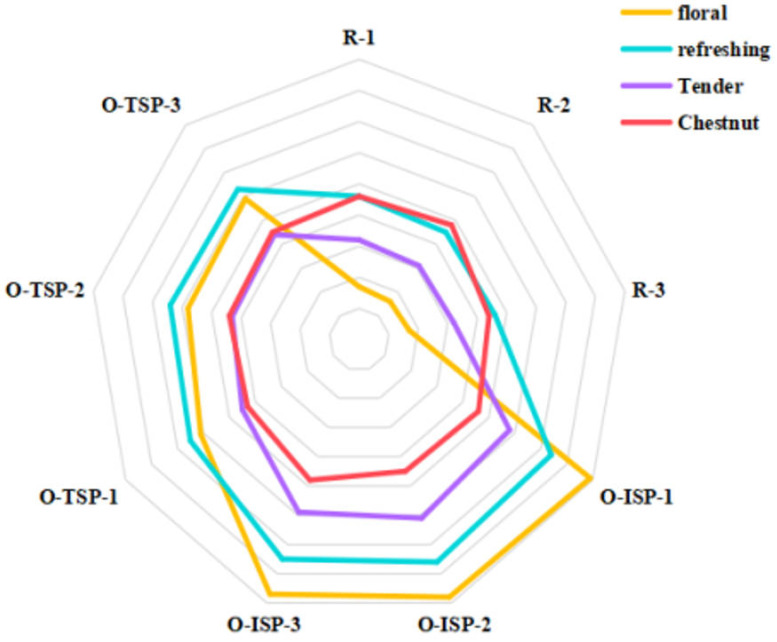
Aroma characteristics of different tea samples, found by sensory evaluation. (Note: R refers to Longjing tea embryo (control treatment); O−TSP refers to osmanthus Longjing tea made by traditional scenting process (non-isolated scenting process); O−ISP refers to osmanthus Longjing tea made by isolated scenting process; The red, purple, yellow and blue line means chestnut fragrance, tender fragrance, floral fragrance and refreshing fragrance).

**Figure 4 foods-13-02985-f004:**
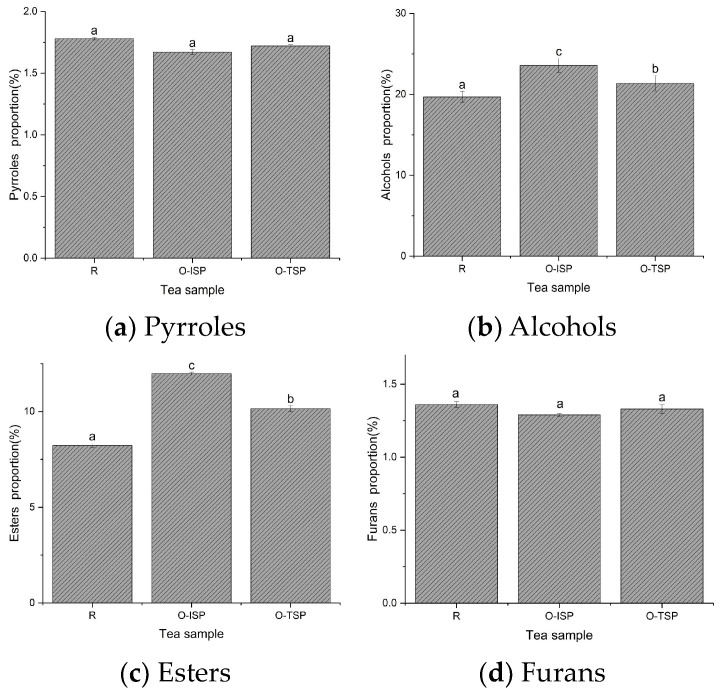
Comparison of aroma fractions with VIP > 1 in Longjing tea for different scenting processes. (**a**) Pyrrole proportions of R, O-ISP, and O-TSP. (**b**) Alcohol proportions of R, O-ISP, and O-TSP. (**c**) Ester proportions of R, O-ISP, and O-TSP. (**d**) Furan proportions of R, O-ISP, and O-TSP. (**e**) Aldehyde proportions of R, O-ISP, and O-TSP. (**f**) Olefin proportions of R, O-ISP, and O-TSP. (**g**) Ketone proportions of R, O-ISP, and O-TSP. (**h**) Acidic proportions of R, O-ISP, and O-TSP. Note: values with different superscripts letters (a–c) in each column indicate significant difference (*p* < 0.05).

**Figure 5 foods-13-02985-f005:**
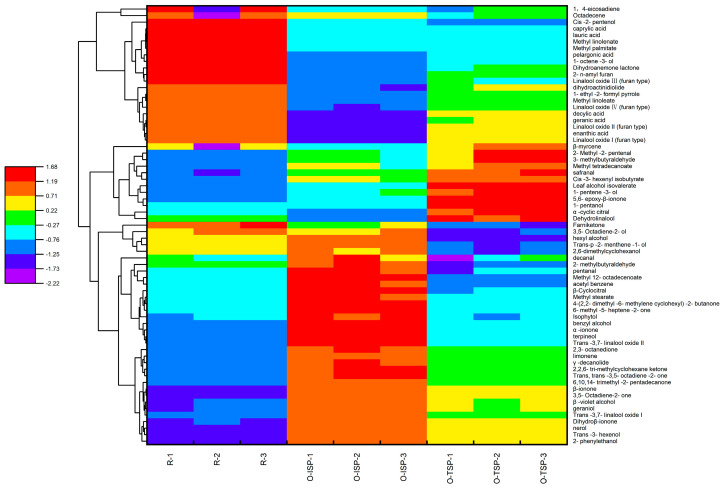
Heat map analysis of aroma compounds in Longjing tea for different scenting processes.

**Figure 6 foods-13-02985-f006:**
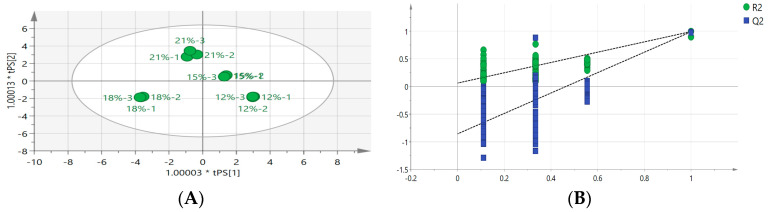
Analysis of volatile compounds of osmanthus Longjing tea for different consumptions of osmanthus. (**A**) The score scatter plot of OPLS-DA for different consumptions of osmanthus. (**B**) Validation of the OPLS-DA model. (**C**) Correlation coefficients of volatile compounds of osmanthus Longjing tea for different consumptions of osmanthus. (**D**) VIP values of volatile compounds of osmanthus Longjing tea for different consumptions of osmanthus.

**Figure 7 foods-13-02985-f007:**
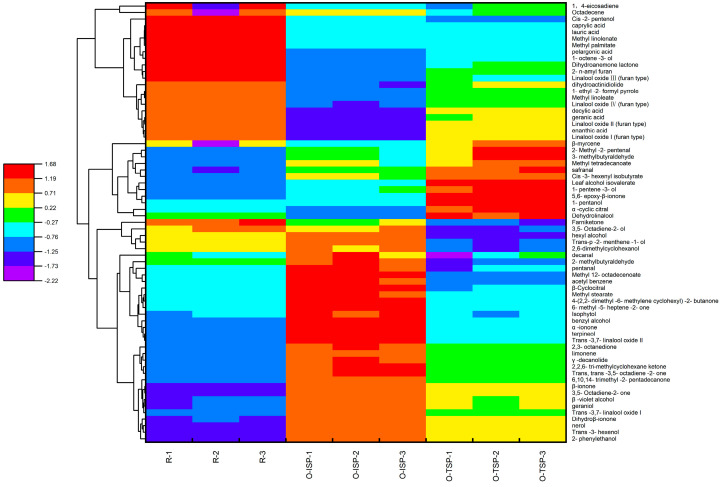
Heat map analysis of aroma compounds under different osmanthus consumption.

**Figure 8 foods-13-02985-f008:**
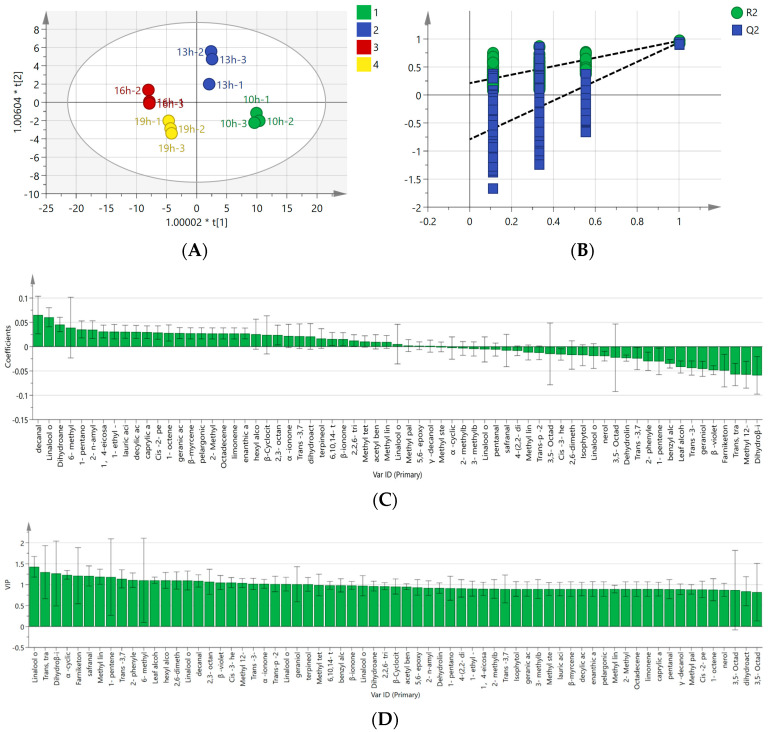
Analysis of volatile compounds of osmanthus Longjing tea for different scenting times. (**A**) The score scatter plots of PCA of different scenting times. (**B**) Validation of the PCA model. (**C**) Correlation coefficients of volatile compounds of osmanthus Longjing tea for different scenting times. (**D**) VIP values of volatile compounds of osmanthus Longjing tea for different scenting times.

**Figure 9 foods-13-02985-f009:**
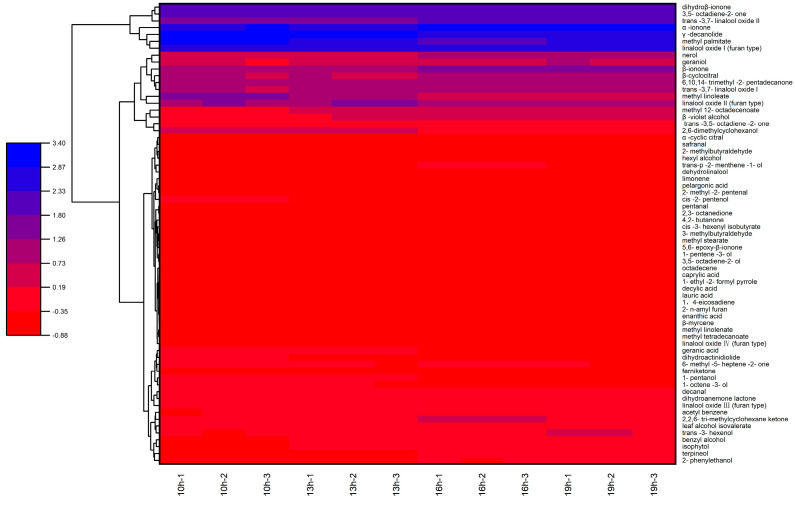
Heat map analysis of aroma compounds under different scenting times.

**Table 1 foods-13-02985-t001:** Identification and quantification of different volatile compounds in Longjing tea for different scenting processes.

No.	Volatile Compounds	RI (Theoretical)	RI (Calculated)	ID ^A^	Relative Concentration (μg/g)		
					R	O-ISP	O-TSP
	Alcohols						
1	2-phenylethanol	1270	1272	MS, RI	-	0.93 ± 0.01 ^b^	0.86 ± 0.02 ^a^
2	2,6-dimethylcyclohexanol	1323	1322	MS, RI	1.47 ± 0.03 ^b^	1.49 ± 0.08 ^b^	1.37 ± 0.03 ^a^
3	linalool oxide I (furan type)	1067	1064	MS, RI	3.87 ± 0.16 ^a^	6.93 ± 0.22 ^c^	5.06 ± 0.21 ^b^
4	linalool oxide II (furan type)	1075	1078	MS, RI	6.53 ± 0.27 ^c^	3.05 ± 0.13 ^a^	5.65 ± 0.19 ^b^
5	linalool oxide III (furan type)	1079	1080	MS, RI	2.18 ± 0.08 ^c^	1.16 ± 0.05 ^a^	1.75 ± 0.06 ^b^
6	linalool oxide IV (furan type)	1092	1094	MS, RI	0.96 ± 0.05 ^c^	0.21 ± 0.00 ^a^	0.68 ± 0.03 ^b^
7	trans-3,7-linalool oxide II	1197	1196	MS, RI	2.55 ± 0.11 ^a^	4.93 ± 0.21 ^b^	3.88 ± 0.13 ^c^
8	trans-3,7-linalool oxide I	1193	1189	MS, RI	1.46 ± 0.03 ^a^	3.95 ± 0.14 ^c^	2.76 ± 0.02 ^b^
9	dehydrolinalool	1190	1088	MS, RI	0.94 ± 0.01 ^b^	0.83 ± 0.00 ^a^	1.12 ± 0.03 ^c^
10	isophytol	2111	2113	MS, RI	1.04 ± 0.04 ^a^	1.96 ± 0.08 ^b^	1.07 ± 0.02 ^a^
11	geraniol	1238	1238	MS, RI	0.43 ± 0.00 ^a^	3.68 ± 0.11 ^c^	2.26 ± 0.09 ^b^
12	1-octene-3-ol	966	968	MS, RI	1.23 ± 0.04 ^c^	0.59 ± 0.02 ^a^	0.83 ± 0.04 ^b^
13	cis-2-pentenol	1621	1625	MS, RI	1.27 ± 0.02 ^b^	0.27 ± 0.00 ^a^	-
14	trans-3-hexenol	822	823	MS, RI	-	1.68 ± 0.08 ^b^	1.33 ± 0.03 ^a^
15	β -violet alcohol	1386	1384	MS, RI	1.50 ± 0.06 ^a^	3.07 ± 0.13 ^c^	1.89 ± 0.05 ^b^
16	1-pentanol	749	749	MS, RI	0.61 ± 0.01 ^a^	0.71 ± 0.02 ^b^	1.92 ± 0.04 ^c^
17	benzyl alcohol	1038	1037	MS, RI	0.98 ± 0.04 ^a^	1.38 ± 0.09 ^b^	1.02 ± 0.01 ^a^
18	trans-p-2-menthene-1-ol	763	768	MS, RI	0.89 ± 0.03 ^b^	0.96 ± 0.04 ^c^	0.72 ± 0.01 ^a^
19	nerol	1231	1230	MS, RI	-	3.27 ± 0.12 ^b^	2.73 ± 0.11 ^a^
20	hexyl alcohol	849	850	MS, RI	0.48 ± 0.01 ^a^	0.69± 0.03 ^b^	-
21	terpineol	1185	1186	MS, RI	-	1.36 ± 0.08 ^b^	0.28 ± 0.00 ^a^
22	3,5-Octadiene-2-ol	1245	1246	MS, RI	0.33 ± 0.00 ^b^	0.32 ± 0.01 ^b^	0.21 ± 0.00 ^a^
23	1-pentene-3-ol	663	665	MS, RI	0.22 ± 0.02 ^a^	0.28 ± 0.01 ^b^	0.41 ± 0.02 ^c^
	Ketones						
1	farniketone	1866	1869	MS, RI	0.86 ± 0.05 ^c^	0.78 ± 0.03 ^b^	0.69 ± 0.02 ^a^
2	3,5-octadiene-2-one	1086	1089	MS, RI	1.23 ± 0.09 ^a^	4.77 ± 0.21 ^c^	4.29 ± 0.15 ^b^
3	trans-3,5-octadiene-2-one	1107	1107	MS, RI	0.43 ± 0.01 ^a^	1.69 ± 0.07 ^c^	1.55 ± 0.04 ^b^
4	dihydroβ-ionone	1436	1438	MS, RI	0.74 ± 0.03 ^a^	5.03 ± 0.35 ^c^	1.71 ± 0.11 ^b^
5	6-methyl-5-heptene-2-one	965	966	MS, RI	-	0.92 ± 0.06	-
6	2,2,6-tri-methylcyclohexane ketone	1267	1269	MS, RI	0.16 ± 0.01 ^a^	2.88 ± 0.08 ^c^	0.69 ± 0.03 ^b^
7	α-ionone	1408	1408	MS, RI	0.96 ± 0.06 ^a^	7.02 ± 0.31 ^c^	2.13 ± 0.09 ^b^
8	β-ionone	1233	1234	MS, RI	0.17 ± 0.01 ^a^	3.96 ± 0.15 ^c^	1.41 ± 0.06 ^b^
9	6,10,14-trimethyl-2-pentadecanone	1260	1263	MS, RI	1.56 ± 0.06 ^a^	3.58 ± 0.13 ^c^	3.17 ± 0.12 ^b^
10	acetyl benzene	1063	1066	MS, RI	1.23 ± 0.02 ^b^	1.66 ± 0.08 ^c^	1.08 ± 0.04 ^a^
11	5,6-epoxy-β-ionone	1478	1475	MS, RI	0.27 ± 0.00 ^a^	0.45 ± 0.02 ^b^	1.61 ± 0.07 ^c^
12	4,2-butanone	846	848	MS, RI	-	0.56 ± 0.02	-
13	2,3-octanedione	966	966	MS, RI	-	0.26 ± 0.01 ^b^	0.13 ± 0.01 ^a^
	Esters						
1	methyl 12-octadecenoate	2072	2070	MS, RI	1.17 ± 0.05 ^a^	2.91 ± 0.09 ^c^	2.04 ± 0.07 ^b^
2	methyl linoleate	2086	2081	MS, RI	3.52 ± 0.23 ^c^	2.03 ± 0.11 ^a^	2.78 ± 0.12 ^b^
3	γ-decanolide	1452	1453	MS, RI	3.98 ± 0.29 ^b^	5.92 ± 0.43 ^c^	3.27 ± 0.25 ^a^
4	methyl palmitate	2243	2243	MS, RI	5.99 ± 0.46 ^b^	5.02 ± 0.38 ^a^	5.03 ± 0.32 ^c^
5	methyl stearate	2112	2113	MS, RI	-	0.37 ± 0.07	-
6	Cis-3-hexenyl isobutyrate	1077	1079	MS, RI	-	0.28 ± 0.02 ^a^	0.43± 0.01 ^b^
7	dihydroactinidiolide	1476	1479	MS, RI	0.88 ± 0.02 ^c^	0.73 ± 0.0 ^a^	0.82 ± 0.07 ^a^
8	dihydroanemone lactone	1507	1508	MS, RI	1.46 ± 0.03 ^c^	1.01 ± 0.03 ^a^	1.15 ± 0.12 ^b^
9	leaf alcohol isovalerate	1301	1302	MS, RI	1.19 ± 0.06 ^a^	1.50 ± 0.09 ^b^	2.35 ± 0.11 ^c^
10	methyl tetradecanoate	1638	1635	MS, RI	-	0.08 ± 0.01 ^a^	0.09 ± 0.00 ^a^
11	methyl linolenate	1097	1099	MS, RI	2.25 ± 0.03 ^c^	0.09 ± 0.00 ^a^	0.11 ± 0.01 ^b^
	Aldehydes						
1	β-cyclocitral	1200	1201	MS, RI	0.04 ± 0.00 ^a^	3.91 ± 0.09 ^a^	0.35 ± 0.02 ^b^
2	decanal	1188	1186	MS, RI	1.00 ± 0.03 ^a^	1.01 ± 0.02 ^a^	0.98 ± 0.02 ^a^
3	safranal	1166	1167	MS, RI	0.50 ± 0.01 ^a^	0.55 ± 0.01 ^b^	0.59 ± 0.02 ^c^
4	α-cyclic citral	1207	1200	MS, RI	0.52 ± 0.02 ^a^	0.47 ± 0.03 ^a^	0.68 ± 0.02 ^b^
5	2-methylbutyraldehyde	638	639	MS, RI	0.59 ± 0.03 ^b^	0.70 ± 0.03 ^a^	0.46 ± 0.01 ^a^
6	3-methylbutyraldehyde	644	641	MS, RI	0.39 ± 0.01 ^a^	0.41 ± 0.00 ^a^	0.43 ± 0.01 ^a^
7	pentanal	689	688	MS, RI	0.18 ± 0.01 ^a^	0.54 ± 0.03 ^b^	0.14 ± 0.00 ^a^
8	2-methyl-2-pentenal	762	766	MS, RI	0.16 ± 0.01 ^a^	0.19 ± 0.00 ^b^	0.22 ± 0.01 ^c^
	Olefins						
1	*β*-myrcene	1169	1168	MS, RI	0.12 ± 0.00 ^b^	0.07 ± 0.00 ^a^	0.13 ± 0.01 ^b^
2	octadecene	963	965	MS, RI	0.15 ± 0.01 ^c^	0.12 ± 0.01 ^b^	0.07 ± 0.01 ^a^
3	limonene	1047	1045	MS, RI	-	0.34 ± 0.01 ^b^	0.15 ± 0.00 ^a^
4	1,4-eicosadiene	1052	1053	MS, RI	0.08 ± 0.00 ^b^	0.02 ± 0.00 ^a^	0.01 ± 0.00 ^a^
	Acids						
1	lauric acid	1557	1558	MS, RI	1.69 ± 0.03 ^b^	0.02 ± 0.00 ^a^	0.05 ± 0.00 ^a^
2	geranic acid	1334	1336	MS, RI	2.58 ± 0.09 ^c^	0.35 ± 0.02 ^a^	1.57 ± 0.08 ^b^
3	pelargonic acid	1302	1304	MS, RI	1.89 ± 0.07 ^c^	0.23 ± 0.00 ^a^	0.55 ± 0.02 ^b^
4	decylic acid	1370	1360	MS, RI	0.78 ± 0.03 ^c^	0.02 ± 0.00 ^a^	0.58 ± 0.01 ^b^
5	caprylic acid	1239	1234	MS, RI	0.69 ± 0.01 ^a^	0.01 ± 0.00 ^a^	0.02 ± 0.00 ^a^
6	enanthic acid	1142	1145	MS, RI	0.79 ± 0.04 ^c^	0.06 ± 0.00 ^a^	0.56 ± 0.02 ^b^
	Pyrroles						
1	1-ethyl-2-formyl pyrrole	896	898	MS, RI	0.96 ± 0.05 ^c^	0.03 ± 0.00 ^a^	0.49 ± 0.02 ^b^
	Furans						
1	2-n-amyl furan	692	695	MS, RI	0.87 ± 0.00 ^c^	0.08 ± 0.00 ^a^	0.36 ± 0.01 ^b^
	Total yield				70.95 ± 0.27 ^a^	106.33 ± 0.85 ^c^	82.75 ± 0.39 ^b^

Mean values ± SDs of four independent experiments are shown. ^a,b,c^ Different letters in the same row indicate significant differences between mean values (*p* < 0.05). - Not detected. ^A^ Identification method. MS: identification based on the NIST mass spectral database; RI: retention index.

## Data Availability

The original contributions presented in the study are included in the article, further inquiries can be directed to the corresponding authors.

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
