# Peer review of "Effect of Isolated Scenting Process on the Aroma Quality of Osmanthus Longjing Tea"

_foods, 2024, doi:10.3390/foods13182985_

Round 1

Reviewer 1 Report

Comments and Suggestions for Authors

The article under the title " Effect of isolated scenting process on the aroma quality of Osmanthus Longjing tea" with the Manuscript ID Foods- 3128803 has the potential to serve as a salient reference within the domain of tea process.

However, authors should respond to the comments below.

1-    The structure of the abstract needs to be improved well. Firstly, the aim/novelty of the study should be given in the abstract and then it should be written in this order: the background of the study, materials and method and then the conclusion and the benefit of the study for the future. The abstract content is quite lacking, and its structure is not suitable for reading. 

2-    Without sharing the results (the results should be shared in discussion part not in introduction, L 89-95) in the introduction section, previous studies using this technique on other teas should be given and the gaps in the literature should be noted. Additionally, the novelty of the study should be stated clearly at the end of the paragraph in the introduction chapter. In other studies, which standard/traditional methods were used for isolation in the other tea types should be cited with a variety of references. Also, what kind of tea has been studied up to now for scenting should be written in the introduction. 

3-    In the material and method section, GC-MS-HPSME should be explained in another sub-chapter not in materials.

4-    Chapters 2.2 and 2.3 in material and method, difference between the scenting techniques traditional and isolation should be explained well using a diagram or a table. Which process/processes are different should be implied clearly in the table. Additionally, chapter 2.3. 2.4, 2.5 should be written with clear sentence and title may be given as the “optimization of …. “

5-    In all subchapters are lacking from discussion. The results should be discussed with another studies for all chapters and what is the importance of this study at the end of the analysis should be explained well.

6-    In Chapter 3.2., for volatile analysis, which compounds are the most significant volatile compounds in the tea samples should be clarified with the content (written with amount/concentration), especially, in esters, olefins and aldehydes which show a significant difference between the samples. these volatile compounds should be discussed in the text and the conclusion.

7-    Correlation analysis may be done between sensorial analysis and volatile compounds to show the effect of remarkable components and their effect on this profile analysis.   

8-    In sensory analysis, explain the lexicons such as refreshing, tender, which components contribute to these sensorial characteristics, and authors should indicate which component is the most present in which example and which character it contributes to its sensorial profile.

9-    In conclusion, the relationship between sensory profile and volatile components should also be written.

10- Grammar and language in general should be checked.

Comments on the Quality of English Language

Editing for English must be done and written in better English. 

Authors should double-check the entire text.  

Author Response

Author's Reply to the Review Report (Reviewer 1)

Comments 1:The structure of the abstract needs to be improved well. Firstly, the aim/novelty of the study should be given in the abstract and then it should be written in this order: the background of the study, materials and method and then the conclusion and the benefit of the study for the future. The abstract content is quite lacking, and its structure is not suitable for reading. 

Response 1:Thank you very much for your constructive suggestions. According to your suggestion, the structure and content of the abstract have been revised in detail.

Comments 2:Without sharing the results (the results should be shared in discussion part not in introduction, L 89-95) in the introduction section, previous studies using this technique on other teas should be given and the gaps in the literature should be noted. Additionally, the novelty of the study should be stated clearly at the end of the paragraph in the introduction chapter. In other studies, which standard/traditional methods were used for isolation in the other tea types should be cited with a variety of references. Also, what kind of tea has been studied up to now for scenting should be written in the introduction.

Response 2:Thank you very much for your constructive suggestions. According to your suggestion, the line 89-95 results have been put on the discussion part. The novelty of the study have been stated clearly at the end of the paragraph in the introduction chapter. The references of scenting methods and aroma quality of other tea types have been cited.

Comments 3:In the material and method section, GC-MS-HPSME should be explained in another sub-chapter not in materials.

Response 3:Thank you very much for your constructive suggestions. According to the suggestion,

the GC-MS-HPSME have been explained in sub-chapter 2.7.1 and 2.7.2.

Comments 4:Chapters 2.2 and 2.3 in material and method, difference between the scenting techniques traditional and isolation should be explained well using a diagram or a table. Which process/processes are different should be implied clearly in the table. Additionally, chapter 2.3. 2.4, 2.5 should be written with clear sentence and title may be given as the “optimization of …. “ 

Response 4:Thank you very much for your constructive suggestions. Using diagrams or tables can clearly illustrate the differences in the manufacturing process. However, due to the limitation of the number of charts in the paper, it was still considered to express it in written form.The wording and names of sections 2.3 and 2.4 have been adjusted to accurately describe the differences in handling. 2.3 and 2.4 mainly focus on studying the effects of blending amount and soaking time on the quality of isolated soaking of osmanthus Longjing tea, rather than optimizing blending amount and soaking time.  

Comments 5:In all subchapters are lacking from discussion. The results should be discussed with another studies for all chapters and what is the importance of this study at the end of the analysis should be explained well.

Response 5:Thank you very much for your constructive suggestions. According to the suggestion, discussion of sub chapters have been added to the manuscript. The importance of this study at the end of the analysis have been explained well.

Comments 6:In Chapter 3.2., for volatile analysis, which compounds are the most significant volatile compounds in the tea samples should be clarified with the content (written with amount/concentration), especially, in esters, olefins and aldehydes which show a significant difference between the samples. these volatile compounds should be discussed in the text and the conclusion.

Response 6:Thank you very much for your constructive suggestions. According to the suggestion, the most significant volatile compounds in the tea samples have been clarified and discussed.

Comments 7:Correlation analysis may be done between sensorial analysis and volatile compounds to show the effect of remarkable components and their effect on this profile analysis.

Response 7:Thank you very much for your constructive suggestions. The suggestion of correlation analysis between sensory analysis and volatile compounds is very good, and I have also considered it. However, due to limitations in the length of the paper and the number of charts, correlation analysis was not conducted.

Comments 8:In sensory analysis, explain the lexicons such as refreshing, tender, which components contribute to these sensorial characteristics, and authors should indicate which component is the most present in which example and which character it contributes to its sensorial profile.

Response 8:Thank you very much for your constructive suggestions. According to the suggestion, Sensory evaluation statements and their impact on quality have been supplemented.

Comments 9:In conclusion, the relationship between sensory profile and volatile components should also be written.

Response 9:Thank you very much for your constructive suggestions. According to the suggestion, the relationship between sensory profile and volatile components have been added to the manuscript.

Comments 10:Grammar and language in general should be checked.

Response 10:Thank you very much for your constructive suggestions. According to the suggestion, we have polished our manuscript carefully and corrected the grammatical, styling, and typos found in the manuscript.

Reviewer 2 Report

Comments and Suggestions for Authors

The authors analyzed the formation of aroma quality in Longjing Osmanthus tea obtained by different aromatization techniques.

The manuscript is interesting. However, we have some issues to explain in the text:

1) Was there any training for the panelists on the specific quality and intensity of the aromas analyzed in the sensory analysis?

2) What references were used by the panelists regarding the quality and intensity of the aromas, including floral, fullness, longevity, and freshness? Please explain in detail in the manuscript the characteristics and description of the intensities and qualities of each of the four aromas analyzed sensorially.

3) Authors must cite the bibliographic references used in the “Sensory evaluation of tea samples”

Author Response

Comments 1:Was there any training for the panelists on the specific quality and intensity of the aromas analyzed in the sensory analysis?

Response 1:Thank you very much for your constructive suggestions. According to your suggestion, the training information for the panelists on the specific quality and intensity of the aromas analyzed in the sensory analysis have been added in detail. The sentence added to the manuscript was "The five sensory evaluation experts were all senior tea appraisers (Grade I) certified by the Ministry of Human Resources and Social Security of China, including three women and two men" .

Comments 2:What references were used by the panelists regarding the quality and intensity of the aromas, including floral, fullness, longevity, and freshness? Please explain in detail in the manuscript the characteristics and description of the intensities and qualities of each of the four aromas analyzed sensorially.

Response 2:Thank you very much for your constructive suggestions. According to your suggestion, the characteristics and description of the intensities and qualities of each of the four aromas analyzed sensorially have been added in sub-chapter 2.6.

    Such as:

    The floral fragrance mainly refers to the smell of tea soup brewed with hot water from scented tea[2]. Floral fragrance was a very important quality parameter for osmanthus Longjing tea, specifically describing the osmanthus fragrance in osmanthu Longjing tea. The fullness  referred to the total amount of aroma and sensory impact. The durability, also known as fragrance retention, referred to the time limit for the aroma of tea soup to remain in a sensory evaluation environment. The freshness referred to the degree of preservation and freshness of the aroma characteristics of tea soup.

    The score represented the intensities among the aroma attributes of tea. The higher the score, the better the aroma intensities , and the higher the overall aroma quality of tea.

Comments 3:Authors must cite the bibliographic references used in the “Sensory evaluation of tea samples”.

Response 3:Thank you very much for your constructive suggestions. According to your suggestion, the bibliographic references used in the “Sensory evaluation of tea samples”  have been cited in the manuscript, which was reference [2]. 

Reviewer 3 Report

Comments and Suggestions for Authors

Revise the abbreviation O-TSP and O- ISP in lines 18 and 21 because both refers to smanthus longjing tea by traditional scenting process.

Refer to the flower tea classification that is mentioned in 50 in order to clarify the idea and sustitute the expression “etc.”

Check the sentence “and triterpe-58 noids of osmanthus have been found that theses have antibacterial activity”, lines 58 and 59.

Add the article “the” to “parts of world” in line 62.

Are lines 100 to 108 part of the abstract due to they do not link to the introduction?

Correct the name of the institute: Research Institute of Hangzhou Academy in line 111.

Change Normal to normal in line 113

Change Dimethoxysilane to dimethoxysilane in line 118

Change Tube Mill Control to tuve mil control in line 122

Correct “which was with slight modified” in line 160

Correct sentence “Each tea infusions was evaluated three times” in line 167

Correct sentence “The retention index (RI) was also be used to identify…” in line 200

It is not necessary to define the tea treatment abbreviations along the manuscritp. You have defined them in line 222 to 226.

Check the composition of lines 260 to 265.

Check abbreviation of tea treatments along the manuscript due to authors used O-ISP to refer to both kind of simples “O-ISP (osmanthus longjing tea by isolated scenting process) and O-ISP (osmanthus longjing tea by traditional scenting process)”

Have these compounds been identified in other kind of flowers?

Why did the authors not use another technique to verify the aroma compounds identified in the tea simples? Techniques such as Kovats index or even databases based on literature references?

Comments on the Quality of English Language

They are indicated in the attached file.

Author Response

Comments 1:Revise the abbreviation O-TSP and O- ISP in lines 18 and 21 because both refers to smanthus longjing tea by traditional scenting process. 

Response 1:Thank you very much for your constructive suggestions. According to your suggestion, the abbreviation O-TSP and O- ISP in lines 18 and 21 have been revised.

Comments 2:Refer to the flower tea classification that is mentioned in 50 in order to clarify the idea and sustitute the expression “etc.”

Response 2:Thank you very much for your constructive suggestions. According to your suggestion, the sentence have been revised. 

Comments 3:Check the sentence “and triterpe-58 noids of osmanthus have been found that theses have antibacterial activity”, lines 58 and 59.

Response 3:Thank you very much for your constructive suggestions. According to your suggestion, the sentence have been checked and revised.

Comments 4:Add the article “the” to “parts of world” in line 62. 

Response 4:Thank you very much for your constructive suggestions. According to your suggestion, the sentence have been revised.  

Comments 5:Are lines 100 to 108 part of the abstract due to they do not link to the introduction?

Response 5:Thank you very much for your constructive suggestions. The lines 100 to 108 is the part of the introduction.

Comments 6:Correct the name of the institute: Research Institute of Hangzhou Academy in line 111.

Response 6:Thank you very much for your constructive suggestions. According to the suggestion, the name of the institute in line 111 have been revised.

Comments 7:Change Normal to normal in line 113

Response 7:Thank you very much for your constructive suggestions. Normal have been revised to normal.

Comments 8:Change Dimethoxysilane to dimethoxysilane in line 118.

Response 8:Thank you very much for your constructive suggestions. According to the suggestion, Dimethoxysilan have been revised to dimethoxysilan.

Comments 9:Change Tube Mill Control to tuve mil control in line 122

Response 9:Thank you very much for your constructive suggestions. According to the suggestion, the Tube Mill Control have been revised.

Comments 10:Correct “which was with slight modified” in line 160

Response 10:Thank you very much for your constructive suggestions. According to the suggestion, the sentence have been revised.

Comments 11:Correct sentence “Each tea infusions was evaluated three times” in line 167

Response 11:Thank you very much for your constructive suggestions. According to the suggestion, the sentence have been revised.

Comments 12:Correct sentence “The retention index (RI) was also be used to identify…” in line 200

Response 12:Thank you very much for your constructive suggestions. According to the suggestion, the sentence have been revised.

Comments 13:It is not necessary to define the tea treatment abbreviations along the manuscript. You have defined them in line 222 to 226.

Response 13:Thank you very much for your constructive suggestions. According to the suggestion, i have corrected it all over the manuscript.

Comments 14:Check the composition of lines 260 to 265.

Response 14:Thank you very much for your constructive suggestions. According to the suggestion, i have corrected it all over the manuscript.

Comments 15:Check abbreviation of tea treatments along the manuscript due to authors used O-ISP to refer to both kind of simples “O-ISP (osmanthus longjing tea by isolated scenting process) and O-ISP (osmanthus longjing tea by traditional scenting process)”

Response 15:Thank you very much for your constructive suggestions. According to the suggestion, i have corrected it all over the manuscript.

Comments 16:Have these compounds been identified in other kind of flowers?

Response 16:No, these compounds do not have been identified in other kind of flowers

Comments 17:Why did the authors not use another technique to verify the aroma compounds identified in the tea simples? Techniques such as Kovats index or even databases based on literature references?

Response 17:Because this method is widely used for the firm analysis of tea aroma compounds. The method is relatively mature. Techniques such as Kovats index or even databases is based on literature references.

Round 2

Reviewer 1 Report

Comments and Suggestions for Authors

Although the authors have corrected some of the submitted comments, there are still deficiencies that need to be addressed.

1-      The authors should refer to the review report where corrections were made, indicating the number of L or the added part (such as which new references added).

2-      The first sentence of Abstract “Scenting was an important…”, this process has a general period, not only in the past, but the sentence should also have started like this “Scenting is an important…”, the authors should be careful in this manner.  

In this example, attention should be paid to all other present tense verbs and these errors should be corrected.

3-       The second sentence, “The aroma quality of osmanthus Longjing tea obtained by different scenting techniques was 12 different”, authors mean general information or their results? Depending on this difference, the tense should be checked. If it refers to the result, it should be given at the end of the abstract.

4-      “The isolated scenting process significantly improved the floral fragrance, refreshing fragrance, tender fragrance of osmanthus Longjing tea” this sentence should be given at the end of the paragraph, also. Hence, the outline of abstract should be reconsidered. The briefly described method (which instrument was used) is still missing in the abstract.

5-      Fig. 2, Fig. 4, Fig. 6, what are the A, B, C, D. The explanation of the figures is missing.

6-      Figure 3 was not cited in the text.

7-      Figure3 and figure 5 have the same legend?

8-      The journal has no restriction on the number of figures and tables.

9-      In conclusion, authors should not cite any references. All citations must be given in the discussion part.

10-  L451-454, Why the authors are going to back and motivation. This is the conclusion part; should not give the lack of information in the literature. The authors should avoid giving these sentences or should write how the results of the study contribute to the literature.

11-  L 452-454, 456-458, “However, research on the 453 impact of isolation fermentation technology on the aroma quality of osmanthus Longjing 454 tea was still relatively weak”.

The authors keep giving the same sentences.

These warnings should be taken very seriously.

12-   Please indicate in which L, authors replied to this comment?

“In conclusion, the relationship between sensory profile and volatile components should also be written.”

13-  Spelling errors continue, a few are presented as examples

L-139, the sentence should not start “With….”,

L-232, discussion,

L-498, In conclusion, the results of This study will provide

Comments on the Quality of English Language

I recommend the authors to reconsider the spelling and grammar. 

Author Response

Comments 1: The authors should refer to the review report where corrections were made, indicating the number of L or the added part (such as which new references added).

Response 1: Thank you very much for your constructive suggestions. According to your suggestion, all changes are in red font, such as the new reference number 35, which can be seen in line 628, 629 and marked with red font.

Comments 2: The first sentence of Abstract “Scenting was an important…”, this process has a general period, not only in the past, but the sentence should also have started like this “Scenting is an important…”, the authors should be careful in this manner.  

In this example, attention should be paid to all other present tense verbs and these errors should be corrected.

Response 2: Thank you very much for your constructive suggestions. According to your suggestion, all the present tense verbs and these errors have been corrected and marked with red font, which can be seen in line 10-35.

Comments 3: The second sentence, “The aroma quality of osmanthus Longjing tea obtained by different scenting techniques was 12 different”, authors mean general information or their results? Depending on this difference, the tense should be checked. If it refers to the result, it should be given at the end of the abstract.

Response 3: Thank you very much for your constructive suggestions. The second sentence have been revised as “There are differences in the aroma quality of osmanthus Longjing tea processed by different scenting processes”, which can be seen in line 11, 12 and marked with red font . It does not refer to the result.

Comments 4: “The isolated scenting process significantly improved the floral fragrance, refreshing fragrance, tender fragrance of osmanthus Longjing tea” this sentence should be given at the end of the paragraph, also. Hence, the outline of abstract should be reconsidered. The briefly described method (which instrument was used) is still missing in the abstract.

Response 4: Thank you very much for your constructive suggestions. According to your suggestion, “The isolated scenting process significantly improved the floral fragrance, refreshing fragrance, tender fragrance of osmanthus Longjing tea” this sentence have been given at the end of the paragraph. And, the outline of abstract have been revised, which can be seen in line 10-35 and marked with red font. The briefly described method (which instrument was used) have been added in the abstract, which can be seen in line 14 and marked with red font.

Comments 5: Fig. 2, Fig. 4, Fig. 6, what are the A, B, C, D. The explanation of the figures is missing.

Response 5: Thank you very much for your constructive suggestions. According to the suggestion, the explanation of A, B, C, D in Fig. 2, Fig. 4, Fig. 6 have been added and revised to another , which can be seen in line 291-295, 362-367, 420-424 and marked with red font..

Comments 6: Figure 3 was not cited in the text.

Response 6: Thank you very much for your constructive suggestions. Figure 3 has been revised to figure 5 in the Line 342, and cited in the Line 315.

Comments 7: Figure3 and figure 5 have the same legend?.

Response 7: Thank you very much for your constructive suggestions. This is a typographical error. It has already been modified, which can be seen in line 342, 407.

Comments 8: The journal has no restriction on the number of figures and tables.

Response 8: Thank you very much for your constructive suggestions. According to the suggestion, the figure 1 and figuer 2 have been added in line 138, 154, which wan mainly to visually represent the difference between the two scenting processes.

Comments 9:  In conclusion, authors should not cite any references. All citations must be given in the discussion part.

Response 9: Thank you very much for your constructive suggestions. According to the suggestion, the references have been cited in conclusion, which can be seen in line 473-481.

Comments 10: L451-454, Why the authors are going to back and motivation. This is the conclusion part; should not give the lack of information in the literature. The authors should avoid giving these sentences or should write how the results of the study contribute to the literature.

Response 10: Thank you very much for your constructive suggestions. According to the suggestion, these sentences in L451-454 have been deleted.

Comments 11: L 452-454, 456-458, “However, research on the 453 impact of isolation fermentation technology on the aroma quality of osmanthus Longjing 454 tea was still relatively weak”. The authors keep giving the same sentences. These warnings should be taken very seriously.

Response 11: Thank you very much for your constructive suggestions. According to the suggestion, these sentences in L452-454, 456-458 have been deleted.

Comments 12: Please indicate in which L, authors replied to this comment?  “In conclusion, the relationship between sensory profile and volatile components should also be written.”

Response 12: Thank you very much for your constructive suggestions. According to the suggestion, we have marked the line where the modification was made. The relationship between sensory profile and volatile components could be seen in line 482-522.

Comments 13: Spelling errors continue, a few are presented as examples.

L-139, the sentence should not start “With….”,

L-232, discussion,

L-498, In conclusion, the results of This study will provide.

Response 13: Thank you very much for your constructive suggestions. According to the suggestion, spelling errors in the whole manuscript have been revised one by one, such as line 10-37, line 50, line 55-58, line 60-61, line 66-72, etc.

Reviewer 2 Report

Comments and Suggestions for Authors

The manuscript is interesting in that it brings new possibilities for applying tools to consumer and food production studies.

Although the manuscript is interesting, the text is confusing and does not provide adequate understanding to the reader.

It does not provide clarity on the importance of so many chromatography methods presented. The reader may not understand what was proposed and the results.

Comments on the Quality of English Language

The quality of the manuscript is poor. It needs an extensive revision.

Author Response

Comments 1:Although the manuscript is interesting, the text is confusing and does not provide adequate understanding to the reader.

Response 1: Thank you very much for your constructive suggestions. According to your suggestion, We have revised the whole article to make it easier for readers to understand the main idea of the study.

Comments 2:It does not provide clarity on the importance of so many chromatography methods presented. The reader may not understand what was proposed and the results.

Response 2: Thank you very much for your constructive suggestions. According to your suggestion, the importance of chromatography methods have been presented the revised manuscript, which can be found in in the beginning of each paragraph in sections 2.7.1 and 2.7.2.
